# Counterfactual History Distillation on Continuous-time Event Sequences

## Abstract

This study aims to distill history events that have essential information for predicting subsequent events with counterfactual analysis. The problem is named Counterfactual History Distillation (CHD). CHD distills a minimum set of events from history, based on which the distribution provided by a trained MTPP model fits the events observed later, and the distribution based on the remaining events in history cannot. It can help understand what event marks may have more influence on the occurrence of future events and what events in history may have a causal relationship with the events observed later. This study proposes a robust solution for CHD, called MTPP-based Counterfactual History Distiller (MTPP-CHD). MTPP-CHD learns to select the optimal event combination from history for the events observed later. Experiment results demonstrate the superiority of MTPP-CHD by outperforming baselines in terms of distillation quality and processing speed.

## 1 Introduction

The Marked Temporal Point Process (MTPP) (Daley & Vere-Jones, 2003) is a well-defined stochastic process that maps historical event sequences to a probability distribution which can be used to predict future events. Learning MTPP by neural networks has been well investigated (Du et al., 2016; Mei & Eisner, 2017; Omi et al., 2019; Zhang et al., 2020a; Zuo et al., 2020; Shchur et al., 2020; Mei et al., 2022; Zhang et al., 2023b; Zhou & Yu, 2023; Lüdke et al., 2023). These algorithms, belonging to the Neural Marked Temporal Point Process (NMTPP) family, enable people to train and use MTPP in high-stake real-world applications like the fake news mitigation (Farajtabar et al., 2017; Zhang et al., 2021b; 2022b) and recommendation systems (Hosseini et al., 2017; Cai et al., 2018).

Counterfactual analysis, also known as counterfactual reasoning, is one of the basic cognitive reasoning approaches. Counterfactual analysis reveals casual relations by searching for the smallest modification to the input that could completely change the output (Tan et al., 2021). For example, to investigate why one piece of disinformation becomes viral on Twitter by counterfactual analysis, we search for the answer by removing retweets of some accounts from the retweet history and then feeding the modified history to an existing model to emulate whether the disinformation still goes viral. If it stopped going viral after we removed retweets of multiple accounts and became viral again when we added them back, we would conclude that these accounts might be the culprits.

Recently, Noorbakhsh & Rodriguez (2022), Zhang et al. (2022b), and Hizli et al. (2023) investigate how the prediction of an MTPP model changes with handcrafted modifications of history with counterfactual analysis. Unlike these studies, we aim to distill a minimum subset of history events with the essential information for predicting the following events using an MTPP model with counterfactual analysis. If the history is modified by removing the minimum subset of events, the accuracy of MTPP model will drop significantly. The problem is named Counterfactual History Distillation (CHD). It can help understand what event marks may have more influence on the occurrence of subsequent events and what events in history may have a causal relationship with the events observed later.

While CHD with conventional counterfactual analysis works in concept, the distilled events are not always satisfactory. It is expected that distilled events have the essential information and the events left in history have trivial information for predicting the subsequent events. However, our study shows this is not true in many scenarios as the prediction accuracy of the MTPP model based on the distilled

events is worse than that based on the events left in history. This means the result of CHD with conventional counterfactual analysis sometimes can be faulty (See Section 2.2 for more information). To address this issue, we refine CHD by adding one more constraint to enforce that distilled events are informative. Without loss of generality, perplexity (Moore & Lewis, 2010) is applied to evaluate prediction accuracy, *i.e.*, how well the distribution of the next events produced by the MTPP model fits the subsequent events observed. CHD is a combinatorial problem. Inspired by the rationalization (Lei et al., 2016), we propose a solution for CHD, named MTPP-based Counterfactual History Distiller (MTPP-CHD), which probes various combinations of events in history with the support of Gumbel-softmax trick (Bengio et al., 2013; Maddison et al., 2017). We show that MTPP-CHD outperforms baseline models in terms of efficiency and distillation quality. We also demonstrate that distilled events can help understand the influence of different marks on the occurrence of future events. In summary, the contributions of this study are threefold:

1. To the best of our knowledge, this study is the first to distill a minimum subset of history events with the essential information for predicting the subsequent events using an MTPP model. We name the problem Counterfactual History Distillation (CHD).

2. This study demonstrates the issues when solving Counterfactual History Distillation (CHD) by conventional counterfactual analysis and refines it with one more constraint to ensure that the distilled events are desirable.

3. This study proposes a robust solution MTPP-CHD for CHD, which learns to select the optimal event combination from history by leveraging Gumbel-softmax trick. Experiment results demonstrate the superiority of MTPP-CHD by outperforming baselines in terms of distillation optimization and processing speed. We also demonstrate that distilled events can help understand the influence of different marks on the occurrence of future events.

## 2 PROBLEM DEFINITION

### 2.1 MARKED TEMPORAL POINT PROCESS

The Marked Temporal Point Process (MTPP) describes a random process of an event sequence $\boldsymbol{x} = (x_1, x_2, \cdots, x_n)$. Each event $x_i = (m_i, t_i)$ comprises a categorical mark $m_i \in \mathbb{M} = \{k_1, k_2, \cdots, k_M\}$ and its occurrence time $t_i$. This paper considers the simple MTPP, which only allows at most one event at every time, thus $t_i < t_j$ if $i < j$. Let $\boldsymbol{\mathcal{H}}_{t_l}$ denote the history up to(include) the time $t_l$ when the most recent event happened and $\boldsymbol{\mathcal{H}}_{t-}$ denote the history up to(exclude) the current time $t$. Given $\boldsymbol{\mathcal{H}}_{t-}$, the conditional intensity function $\lambda^*(m, t)$ is the probability that an event with mark $m$ will happen at time $t$ (Daley & Vere-Jones, 2003)[1]:

$$\lambda^*(m, t) = \lim_{\Delta t \to 0} \frac{P(m, t \in (t, t + \Delta t] | \boldsymbol{\mathcal{H}}_{t-})}{\Delta t}. \tag{1}$$

With $\lambda^*(m, t)$, we can define the joint probability distribution $p^*(m, t)$ of the next event whose mark is $m$ and the time to occur is $t$.

$$p^*(m, t) = \lambda^*(m, t) \exp\left(-\sum_{k \in \mathbb{M}} \int_{t_l}^{t} \lambda^*(k, \tau) \mathrm{d}\tau\right). \tag{2}$$

The Negative Log-Likelihood (NLL) loss on $\boldsymbol{x}$ observed in a time interval $[t_0, T]$ is:

$$L = -\log p(\boldsymbol{x}) = -\sum_{i=1}^{n} \log \lambda^*(m_i, t_i) + \sum_{k \in \mathbb{M}} \int_{t_0}^{T} \lambda^*(k, \tau) \mathrm{d}\tau. \tag{3}$$

Equation (3) is the training loss of many MTPP models (Du et al., 2016; Mei & Eisner, 2017; Omi et al., 2019; Zhang et al., 2020a; Zuo et al., 2020; Shchur et al., 2020; Mei et al., 2022).

### 2.2 PROBLEM STATEMENT AND FORMULATION

A dataset $\boldsymbol{D}$ contains event sequences. Suppose an MTPP model has been trained on $\boldsymbol{D}$. For any subsequence $(x_1, \cdots, x_j, x_{j+1}, \cdots, x_{n-1}, x_n)$ of an event sequence in $\boldsymbol{D}$, the first part $(x_1, \cdots, x_j)$,

---

[1]The asterisk reminds that this function conditions on history.

Table 1: When solving CHD with conventional counterfactual analysis, the percentage of $\mathcal{H}_d$s that have less information than the corresponding $\mathcal{H}_l$s.

| StackOverflow ($|\mathbf{x}_o| = 15, |\mathcal{H}_f| = 40$) | Retweet ($|\mathbf{x}_o| = 10, |\mathcal{H}_f| = 25$) | Yelp ($|\mathbf{x}_o| = 10, |\mathcal{H}_f| = 25$) |
|---|---|---|
| 19.068 % | 0.4627 % | 1.0114 % |

denoted as $\mathcal{H}_f$, is the history relative to the second part $(x_{j+1}, \cdots, x_{n-1}, x_n)$, denoted as $\boldsymbol{x}_o$. The second part consists of events observed after the first part.

For each $x_i \in \boldsymbol{x}_o$, the MTPP model can be used to produce the distribution of the next event $p(x|\mathcal{H})$ where $\mathcal{H}$ includes the previous events before $x_i$ in the subsequence. We can evaluate how well the distribution fits $x_i$ by using $p(x_i|\mathcal{H})$. The high $p(x_i|\mathcal{H})$ indicates fitting well. $\mathcal{H}$ consists of events in $\mathcal{H}_f$ and events in $\boldsymbol{x}_o$ before $x_i$. We aim to search for a subset of the events in $\mathcal{H}_f$. For different subsets, the events in $\boldsymbol{x}_o$ before $x_i$ are same. To make the presentation simple in the rest of the paper, $\mathcal{H}$ represents the events in $\mathcal{H}_f$ and ignores the events in $\boldsymbol{x}_o$. To judge if $\boldsymbol{x}_o$ fits $p(\boldsymbol{x}_o|\mathcal{H})$, we use *perplexity*, denoted as $\mathrm{ppl}(p(\boldsymbol{x}_o|\mathcal{H}))$. Its definition is:

$$\mathrm{ppl}(p(\boldsymbol{x}_o|\mathcal{H})) = \exp\left(-\frac{1}{|\boldsymbol{x}_o|} \log \prod_{x_i \in \boldsymbol{x}_o} p(x_i|\mathcal{H})\right). \tag{4}$$

A lower perplexity indicates $p(\boldsymbol{x}_o|\mathcal{H})$ better fitting $\boldsymbol{x}_o$. Perplexity has been widely used to select in-domain data from non-specific-domain datasets (Moore & Lewis, 2010; Toral et al., 2015; Feng et al., 2022) and evaluation of Large Language Models (LLMs) (Brown et al., 2020; Du et al., 2022; Zhang et al., 2022a; Zeng et al., 2022).

Counterfactual History Distillation (CHD) aims to distill essential events in $\mathcal{H}_f$ that enable the MTPP model to generate $p(\boldsymbol{x}_o|\mathcal{H}_f)$ fitting $\boldsymbol{x}_o$. Following conventional counterfactual analysis, the problem is to identify the minimum subset $\mathcal{H}_d \subseteq \mathcal{H}_f$ so that $p(\boldsymbol{x}_o|\mathcal{H}_f)$ fits $\boldsymbol{x}_o$, but $p(\boldsymbol{x}_o|\mathcal{H}_l = \mathcal{H}_f - \mathcal{H}_d)$ does not. The formal definition is:

$$\min_{\mathcal{H}_d \subseteq \mathcal{H}_f} |\mathcal{H}_d|$$
$$\text{s.t.} \quad \frac{\mathrm{ppl}(p(\boldsymbol{x}_o|\mathcal{H}_f))}{\mathrm{ppl}(p(\boldsymbol{x}_o|\mathcal{H}_l))} \leqslant \epsilon_l \tag{5}$$

where $\epsilon_l \in (0, 1)$ is a threshold to ensure the information in $\mathcal{H}_l$ about $\boldsymbol{x}_o$ is trivial.

However, our study shows that the result of the conventional counterfactual analysis is problematic in many scenarios. Table 1 shows the percentage of subsequences in three real-world datasets where $\mathrm{ppl}(p(\boldsymbol{x}_o|\mathcal{H}_d))$ is greater than $\mathrm{ppl}(p(\boldsymbol{x}_o|\mathcal{H}_l))$ by solving the optimization problem in Equation (5)[2]. This means $\mathcal{H}_l$ sometimes contains more information about $\boldsymbol{x}_o$ than $\mathcal{H}_d$, which is undesirable. To address this issue, we refine CHD by adding one more constraint to enforce that the information in $\mathcal{H}_d$ is significantly more than $\mathcal{H}_l$ for predicting $\boldsymbol{x}_o$.

$$\min_{\mathcal{H}_d \subseteq \mathcal{H}_f} |\mathcal{H}_d|$$
$$\text{s.t.} \quad \frac{\mathrm{ppl}(p(\boldsymbol{x}_o|\mathcal{H}_f))}{\mathrm{ppl}(p(\boldsymbol{x}_o|\mathcal{H}_l))} \leqslant \epsilon_l,$$
$$\frac{\mathrm{ppl}(p(\boldsymbol{x}_o|\mathcal{H}_f))}{\mathrm{ppl}(p(\boldsymbol{x}_o|\mathcal{H}_d))} \geqslant \epsilon_d. \tag{6}$$

where $\epsilon_d \in (0, 1)$ is another threshold to ensure the information in $\mathcal{H}_d$ about $\boldsymbol{x}_o$ is sufficient. $\epsilon_d > \epsilon_l$. For the ease of computation, we apply logarithm to the constraints in Equation (6):

$$\min_{\mathcal{H}_d \subseteq \mathcal{H}_f} |\mathcal{H}_d|$$
$$\text{s.t.} \quad \log\mathrm{ppl}(p(\boldsymbol{x}_o|\mathcal{H}_f)) - \log\mathrm{ppl}(p(\boldsymbol{x}_o|\mathcal{H}_l)) \leqslant \log\epsilon_l,$$
$$\log\mathrm{ppl}(p(\boldsymbol{x}_o|\mathcal{H}_f)) - \log\mathrm{ppl}(p(\boldsymbol{x}_o|\mathcal{H}_d)) \geqslant \log\epsilon_d. \tag{7}$$

---

[2]Here, we solve the optimization problem by training a MTPP-CHD with $L_n$ and $L_l$. See Section 3 for definitions of MTPP-CHD, $L_n$, and $L_l$.

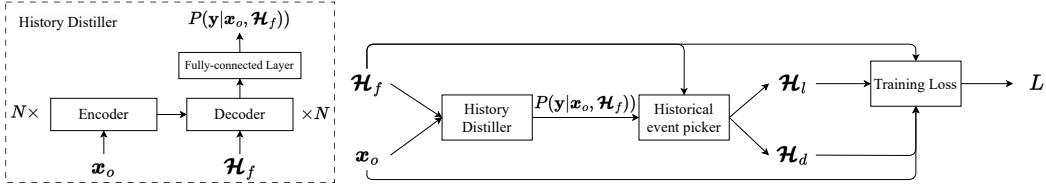

Figure 1: Architecture of MTPP-CHD.

The perplexity of $p(\boldsymbol{x}_o|\mathcal{H})$ is tricky if $\mathcal{H}$ is an empty set $\varnothing$. In this case, we have $p(\boldsymbol{x}_o|\varnothing) = p(\boldsymbol{x}_o, \varnothing)/p(\varnothing)$ where $p(\varnothing) = 0$. Due to division by 0, $p(\boldsymbol{x}_o|\varnothing)$ is undefined. Intuitively, when we continuously remove events from $\mathcal{H}$, the information in $\mathcal{H}$ decreases so that $p(\boldsymbol{x}_o|\mathcal{H})$ approaches zero. So, we define $p(\boldsymbol{x}_o|\varnothing)$ an infinitesimal number, which induces $\mathrm{ppl}(p(\boldsymbol{x}_o|\varnothing)) \to +\infty$. We have the following proposition (proven in Appendix A.1):

**Proposition 1.** *Counterfactual History Distillation (CHD) defined in Equation* (7) *always has a solution for any* $\epsilon_l \in (0, 1)$, $\epsilon_d \in (0, 1)$, *and* $\epsilon_d > \epsilon_l$.

## 3 MTPP-BASED COUNTERFACTUAL HISTORY DISTILLER (MTPP-CHD)

The proposed CHD solution, MTPP-based Counterfactual History Distiller (MTPP-CHD), is sketched in Figure 1. MTPP-CHD consists of three components. The first component, *history distiller*, processes $\mathcal{H}_f$ and $\boldsymbol{x}_o$ using an encoder-decoder transformer, then pushes the resultant representations into a fully connected layer. The output is $p(\mathbf{y}|\mathcal{H}_f, \boldsymbol{x}_o)$. Here, $\mathbf{y}$ is a mask vector of size $|\mathcal{H}_f|$, each for one event in $\mathcal{H}_f$. For $y_i \in \mathbf{y}$, if $y_i = 0$, the corresponding element $x_i \in \mathcal{H}_f$ goes to $\mathcal{H}_l$. If $y_i = 1$, the corresponding element $x_i \in \mathcal{H}_f$ goes to $\mathcal{H}_d$. All trainable parameters are in the first component. The second component, *historical event picker*, derives $\mathcal{H}_l$ and $\mathcal{H}_d$ based on $p(\mathbf{y}|\mathcal{H}_f, \boldsymbol{x}_o)$. The third component, *training loss*, employs a trained MTPP model to evaluate the derived $\mathcal{H}_l$ and $\mathcal{H}_d$ for training MTPP-CHD. The third component only exists during training.

### 3.1 TRAINING OF MTPP-CHD

Training MTPP-CHD begins by initializing the parameters of the history distiller. Given history $\mathcal{H}_f$ and $\boldsymbol{x}_o$, the history distiller processes them using an encoder-decoder transformer to represent each event in $\mathcal{H}_f$ so that it is aware of other events in $\mathcal{H}_f$ and events in $\boldsymbol{x}_o$. Then, the representations of events in $\mathcal{H}_f$ are fed to a fully connected layer and the output is $p(\mathbf{y}|\mathcal{H}_f, \boldsymbol{x}_o)$, the distribution of mask vector $\mathbf{y}$. For $y_i \in \mathbf{y}$, $p(y_i|\mathcal{H}_f, \boldsymbol{x}_o)$ is a categorical distribution of two categories, *i.e.*, $\{0, 1\}$. If $p(y_i = 0|\mathcal{H}_f, \boldsymbol{x}_o)$ is larger, the corresponding event is more likely to go to $\mathcal{H}_l$. If $p(y_i = 1|\mathcal{H}_f, \boldsymbol{x}_o)$ is larger, the corresponding event is more likely to go to $\mathcal{H}_d$.

Next, the historical event picker samples masks from $p(\mathbf{y}|\mathcal{H}_f, \boldsymbol{x}_o)$ and obtains the corresponding $\mathcal{H}_d$ and $\mathcal{H}_l$ for multiple times. Algorithm 1 shows how a sample, denoted as $\hat{\mathbf{y}}$, is drawn and processed to return $\mathcal{H}_d$ and $\mathcal{H}_l$ during training.

**Algorithm 1** Historical event picker during training.

---
**Input:** $\mathcal{H}_f$ and $p(\mathbf{y}|\boldsymbol{x}_o, \mathcal{H}_f)$;
**Output:** $\mathcal{H}_l$ and $\mathcal{H}_d$;
$\mathcal{H}_d = \varnothing$, $\mathcal{H}_l = \varnothing$;
$\hat{\mathbf{y}} \leftarrow$ sampling $p(\mathbf{y}|\boldsymbol{x}_o, \mathcal{H}_f)$ with Gumbel-softmax trick;
**for** $x_i \in \mathcal{H}_f$ **do**
  **if** $\hat{y}_i == 1$ **then**
    $\mathcal{H}_d \leftarrow \mathcal{H}_d \cup x_i$;
  **else**
    $\mathcal{H}_l \leftarrow \mathcal{H}_l \cup x_i$;
  **end if**
**end for**
**return** $\mathcal{H}_l, \mathcal{H}_d$;

---

Specially, a sample is drawn by sampling categorical distribution $p(y_i|\mathcal{H}_f, \boldsymbol{x}_o)$ for each element $y_i \in \mathbf{y}$. If $\hat{y}_i = 0$, the corresponding element in $\mathcal{H}_f$ goes to $\mathcal{H}_l$. If $\hat{y}_i = 1$, the corresponding element in $\mathcal{H}_f$ goes to $\mathcal{H}_d$. To draw sample from $p(y_i|\mathcal{H}_f, \boldsymbol{x}_o)$ in a differentiable way, we use the Gumbel-softmax trick (Bengio et al., 2013; Maddison et al., 2017). After the sample $\hat{\mathbf{y}}$ is drawn, the distilled events form $\mathcal{H}_d$ and the remaining events constitute $\mathcal{H}_l$. In natural language processing, a similar method has been used for rationalization (Lei et al., 2016) to search a document for an optimal combination of sentences related to a claim.

The third component evaluates $\mathcal{H}_d$ and $\mathcal{H}_l$ for the loss. According to Equation (7), the loss function of MTPP-CHD comprises two aspects: $L_e$ for enforcing perplexity-based constraints and $L_n$ for

minimizing the length of $\mathcal{H}_d$. With $\mathcal{H}_d$ and $\mathcal{H}_l$ derived from sample $\hat{\mathbf{y}}$, we enforce perplexity-based constraints in a differentiable way by using two surrogate hinge losses, inspired by (Mothilal et al., 2020; Tan et al., 2021):

$$L_l(\hat{\mathbf{y}}) = \max(\log \mathrm{ppl}(p(\boldsymbol{x}_o|\mathcal{H}_f)) - \log \mathrm{ppl}(p(\boldsymbol{x}_o|\mathcal{H}_l)) - \log \epsilon_l, 0).$$
$$L_d(\hat{\mathbf{y}}) = \max(\log \mathrm{ppl}(p(\boldsymbol{x}_o|\mathcal{H}_d)) - \log \mathrm{ppl}(p(\boldsymbol{x}_o|\mathcal{H}_f)) + \log \epsilon_d, 0).$$

(8)

Reducing loss $L_l$ will increase $\log \mathrm{ppl}(p(\boldsymbol{x}_o|\mathcal{H}_l))$ until its gap to $\log \mathrm{ppl}(p(\boldsymbol{x}_o|\mathcal{H}_f))$ is larger than $\epsilon_l$. Reducing loss $L_d$ will decrease $\log \mathrm{ppl}(p(\boldsymbol{x}_o|\mathcal{H}_d))$ until its gap to $\log \mathrm{ppl}(p(\boldsymbol{x}_o|\mathcal{H}_f))$ is smaller than $\log \epsilon_d$. The loss $L_e$ is based on $N$ samples from $p(\mathbf{y}|\mathcal{H}_f, \boldsymbol{x}_o)$, where $\hat{\mathbf{y}}_i$ refers to the $i$-th sample:

$$L_e = \mathbb{E}_{\hat{\mathbf{y}} \sim p(\mathbf{y}|\mathcal{H}_f, \boldsymbol{x}_o)}(L_l(\hat{\mathbf{y}}) + L_d(\hat{\mathbf{y}}))$$
$$\approx \frac{1}{N} \sum_{i=1}^{N}(L_l(\hat{\mathbf{y}}_i) + L_d(\hat{\mathbf{y}}_i).$$

(9)

We use a trained MTPP model to estimate the conditional probability distribution $p(\boldsymbol{x}_o|\mathcal{H})$ in Equation (8). Any MTPP models outputting $p^*(m, t)$ defined in Equation (2) should work. The only requirement is that the MTPP model is differentiable, so MTPP-CHD obtains the gradient $\nabla_{\hat{\mathbf{y}}} L_e$ to enable training. In this paper, the trained MTPP model is FullyNN (Omi et al., 2019). Details about FullyNN and how we train FullyNN on $\boldsymbol{D}$ are available in Appendix B.2.

The loss $L_n$ aims to minimize the length of $\mathcal{H}_d$, *i.e.*, the number of distilled events. Because $\mathcal{H}_d$ is derived from $\hat{\mathbf{y}}$, minimizing the length of $\mathcal{H}_d$ equals to maximizing $\ell^0$-norm of $\hat{\mathbf{y}}$, *i.e.*, the number of nonzero elements in $\hat{\mathbf{y}}$. However, the $\ell^0$-norm is not differentiable. As a workaround, some studies optimize the differentiable $\ell^1$-norm (Tan et al., 2021). However, optimizing $\ell^1$-norm of a vector $\mathbf{a} \in \mathbb{R}^d$ has limited effects on optimizing $\ell^0$-norm because there is no consistent relation between them. $\ell^0$-norm can decrease, stay unchanged, or even increase when $\ell^1$-norm decreases. Interestingly, $\ell^1$-norm of $\hat{\mathbf{y}}$ has a consistent relation with $\ell^0$-norm of $\hat{\mathbf{y}}$ because $\hat{\mathbf{y}}$ only contains 0 and 1. This means that $\ell^0$-norm is always equal to $\ell^1$-norm for $\hat{\mathbf{y}}$. This means optimizing $\hat{\mathbf{y}}$'s $\ell^1$-norm is equivalent to optimizing $\hat{\mathbf{y}}$'s $\ell^0$-norm. We define $L_n$ as the normalized $\ell^1$-norm by dividing the length of $\hat{\mathbf{y}}$:

$$L_n = \frac{\|\hat{\mathbf{y}}\|_1}{|\hat{\mathbf{y}}|}.$$

(10)

With $L_e$ and $L_n$ properly defined, the training loss $L$ of MTPP-CHD is the sum of $L_n$ and $L_e$. We use two hyperparameters $\alpha$ and $\beta$ to balance the number of distilled events and the perplexity gap.

$$L = \alpha L_n + \beta L_e.$$

(11)

## 3.2 INFERENCE OF MTPP-CHD

Counterfactual history distillation with the learned MTPP-CHD consists of the trained history distiller and an inference-specific historical event picker. The inference process is presented in Algorithm 2. The history distiller takes in history $\mathcal{H}_f$ and $\boldsymbol{x}_o$ for $p(\mathbf{y}|\mathcal{H}_f, \boldsymbol{x}_o)$. During inference, the historical event picker returns the optimal $\mathcal{H}_d$ based on $p(\mathbf{y}|\mathcal{H}_f, \boldsymbol{x}_o)$. To do that, elements $y_i \in \mathbf{y}$ are sorted in descending order based on $p(y_i = 1|\mathcal{H}_f, \boldsymbol{x}_o)$. Initially, $\mathcal{H}_d$ is empty and $k = 1$. The top-$k$ element is moved to $\mathcal{H}_d$ and $\mathcal{H}_l$ includes the remaining elements. With the trained MTPP model, $\mathrm{ppl}(p(\boldsymbol{x}_o|\mathcal{H}_l))$ and $\mathrm{ppl}(p(\boldsymbol{x}_o|\mathcal{H}_d))$ are calculated. If the constraints in Equation (7) are satisfied, $\mathcal{H}_d$ is returned. If not, $k = k + 1$ and the same process is taken until the constraints in Equation (7) are satisfied and $\mathcal{H}_d$ is returned.

**Algorithm 2** Historical event picker during inference.

---

**Input:** $\mathcal{H}_f$ and $p(\mathbf{y}|\boldsymbol{x}_o, \mathcal{H}_f)$;
**Output:** $\mathcal{H}_d$;
$\mathcal{H}_f^* \leftarrow$ sort $\mathcal{H}_f$ in descending order of $p(y_i|\boldsymbol{x}_o, \mathcal{H}_f)$;
$\mathcal{H}_d = \varnothing, \mathcal{H}_l = \mathcal{H}_f$;
**for** $x_i$ in $\mathcal{H}_f^*$ **do**
  **if** $\mathcal{H}_d, \mathcal{H}_l$ satisfy the constraints in Equation (7) **then**
    break;
  **end if**
  $\mathcal{H}_d \leftarrow \mathcal{H}_d \cup x_i$;
  $\mathcal{H}_l \leftarrow \mathcal{H}_l - x_i$;
**end for**
**return** $\mathcal{H}_d$;

---

## 4 EXPERIMENTS

This section evaluates the effectiveness of MTPP-CHD by answering following questions: (i) Does solving CHD in Equation (7) lead to better distillation compared with Equation (5)? (ii) Does the

proposed MTPP-CHD solve CHD with $L_n$ and $L_e$ with good distillation quality and efficiency? , and (iii) What statistical features or knowledge can be exploited from the $\mathcal{H}_d$?

## 4.1 Experiment settings

The same experiments are run 3 times with different random seeds, and their mean and standard deviation (1-sigma) are reported. More details are available in Appendix B.2 including the hardware and software for the experiments, the hyperparameters of MTPP-CHD, the setting of $\epsilon_l$ and $\epsilon_d$, and a brief introduction of FullyNN.

**Baseline Models** To our knowledge, no previous studies investigated CHD in the context of MTPP. This means we do not have baselines from existing studies to compare with. Brute force is infeasible because solving a combinatorial problem like CHD is NP-hard (Karp, 1972). We notice some studies applying counterfactual analysis in recommender systems. They greedily search for the smallest subset of history that the recommendation would change with the subset removed (Ghazimatin et al., 2020; Tran et al., 2021; Zhong & Negre, 2022). This motivates us to adopt a Greedy Search (GS) baseline. It solves CHD by incrementally selecting from $\mathcal{H}_f$ the event that increases the gap between $\log \mathrm{ppl}(\boldsymbol{x}_o|\mathcal{H}_l)$ and $\log \mathrm{ppl}(\boldsymbol{x}_o|\mathcal{H}_d)$ the most and inserting it to $\mathcal{H}_d$ until the two constraints are satisfied. We also take a Random Distillation (RD) baseline to show the difficulty of CHD. RD randomly moves $Q$ events from $\mathcal{H}_f$ to $\mathcal{H}_d$ and calculates the gap between $\log \mathrm{ppl}(\boldsymbol{x}_o|\mathcal{H}_f)$ and $\log \mathrm{ppl}(\boldsymbol{x}_o|\mathcal{H}_l)$. This is repeated multiple times and the average of these gaps is recorded. $Q$ starts from 0. RD stops and returns $Q$ when the average gaps satisfy the two constraints in Equation (7); otherwise, increase $Q$ by 1 and repeat the previous process.

**Evaluation Metrics** We are concerned to which extent the optimization objective of CHD is achieved, *i.e.*, minimizing $|\mathcal{H}_d|$ while two constraints are satisfied. For all $(\mathcal{H}_f, \boldsymbol{x}_o)$ pairs in the test dataset $\boldsymbol{T}$, we calculate the average length of $\mathcal{H}_d$ provided by a CHD approach.

$$|\bar{\mathcal{H}}_d| = \frac{1}{|\boldsymbol{T}|} \sum_{(\mathcal{H}_f, \boldsymbol{x}_o) \in \boldsymbol{T}} |\mathcal{H}_d|. \tag{12}$$

Lower $|\bar{\mathcal{H}}_d|$ indicates the CHD approach obtains shorter $\mathcal{H}_d$ that meets the constraints in Equation (7), thus better.

**Datasets** We test MTPP-CHD and baselines on three real-world datasets: Retweet (Zhao et al., 2015), StackOverflow (Leskovec & Krevl, 2014) and Yelp. Retweet contains 2.6 million events, StackOverflow 480K events, and Yelp 400K events. All subsequences with $n = |\mathcal{H}_f| + |\boldsymbol{x}_o|$ events are extracted from these datasets. Further, each subsequence is split into $\mathcal{H}_f$ and $\boldsymbol{x}_o$. Each dataset has 5 different $|\mathcal{H}_f|$ and $|\boldsymbol{x}_o|$ settings. More details are presented in Appendix B.

## 4.2 Experiment Results

### 4.2.1 Effectiveness of Counterfactual Analysis Refinement

CHD can be tackled by working out the optimization problem defined in Equation (5). This method is based on counterfactual analysis but is problematic as pointed out in Section 2.2. To prevent such undesirable results, we refine the counterfactual analysis with a new constraint on $\mathcal{H}_d$ as defined in Equation (7). To investigate the impact of the new constraint, we compare MTPP-CHD, our solution of CHD based on the counterfactual analysis with $\mathcal{H}_d$ constraint, against MTPP-CHD without refinement, based on counterfactual analysis without $\mathcal{H}_d$ constraint. Figure 2 presents the distribution of $\log \mathrm{ppl}(p(\boldsymbol{x}_o|\mathcal{H}_l)) - \log \mathrm{ppl}(p(\boldsymbol{x}_o|\mathcal{H}_d))$ of our MTPP-CHD and the MTPP-CHD without refinement. If $\mathcal{H}_d$ has less information than $\mathcal{H}_l$, the value of $\log \mathrm{ppl}(p(\boldsymbol{x}_o|\mathcal{H}_l)) - \log \mathrm{ppl}(p(\boldsymbol{x}_o|\mathcal{H}_d))$ is less than zero; otherwise greater than 0. Our MTPP-CHD demonstrates the resultant $\mathcal{H}_d$ always has more information than the corresponding $\mathcal{H}_l$. In contrast, MTPP-CHD without refinement may lead to some resultant $\mathcal{H}_d$s having less information than corresponding $\mathcal{H}_l$s. Such an undesirable situation is significant on StackOverflow.

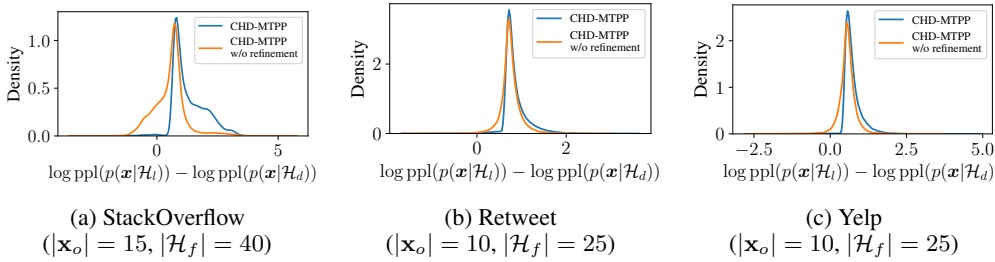

Figure 2: The distribution of $\log \mathrm{ppl}(p(\boldsymbol{x}_o|\boldsymbol{\mathcal{H}}_l)) - \log \mathrm{ppl}(p(\boldsymbol{x}_o|\boldsymbol{\mathcal{H}}_d))$ of our MTPP-CHD and the MTPP-CHD without refinement.

### 4.2.2 DISTILLATION QUALITY

We solve CHD by working out the optimization problem defined in Equation (7), where the optimization objective is to identify $\boldsymbol{\mathcal{H}}_d$ with the minimum number of events under two constraints. The resultant $\boldsymbol{\mathcal{H}}_d$ with fewer events indicates a better solution. Table 2 reports $|\bar{\boldsymbol{\mathcal{H}}}_d|$ using our MTPP-CHD and baselines. First, GS outperforms RD by a consistent and noticeable margin on all datasets. It demonstrates that CHD is a difficult task that cannot be properly solved with a simple solution like RD. Second, our MTPP-CHD demonstrates the performance better than both baselines. With GS, it repeatedly identifies the individual event that affects $L_e$ the most and moves it from $\boldsymbol{\mathcal{H}}_f$ to $\boldsymbol{\mathcal{H}}_d$. However, this method cannot capture the effect of event combinations in $\boldsymbol{\mathcal{H}}_f$ and may lead to suboptimal solutions. In contrast, our MTPP-CHD overcomes the weakness of GS by searching for optimal event combinations and therefore demonstrates better performance.

Table 2: The average length of $\boldsymbol{\mathcal{H}}_d$ returned by MTPP-CHD and baselines (the standard deviation of GS is 0 because GS is deterministic).

|  | $|\mathbf{x}_o|$ | $|\boldsymbol{\mathcal{H}}_f|$ | MTPP-CHD | GS | RD |
|---|---|---|---|---|---|
| StackOverflow | 15 | 40 | $\mathbf{21.484}_{\pm\mathbf{0.0073}}$ | $23.681_{\pm 0.0000}$ | $36.424_{\pm 0.0033}$ |
|  | 15 | 45 | $\mathbf{23.700}_{\pm\mathbf{0.0802}}$ | $25.700_{\pm 0.0000}$ | $40.582_{\pm 0.0042}$ |
|  | 15 | 50 | $\mathbf{26.115}_{\pm\mathbf{0.5226}}$ | $27.699_{\pm 0.0000}$ | $44.693_{\pm 0.0015}$ |
|  | 20 | 50 | $\mathbf{27.416}_{\pm\mathbf{0.0974}}$ | $28.927_{\pm 0.0000}$ | $44.898_{\pm 0.0046}$ |
|  | 25 | 50 | $\mathbf{27.811}_{\pm\mathbf{0.2973}}$ | $29.636_{\pm 0.0000}$ | $45.159_{\pm 0.0011}$ |
| Retweet | 10 | 25 | $\mathbf{12.281}_{\pm\mathbf{0.2001}}$ | $14.722_{\pm 0.0000}$ | $24.004_{\pm 0.0004}$ |
|  | 10 | 30 | $\mathbf{13.297}_{\pm\mathbf{0.2264}}$ | $16.511_{\pm 0.0000}$ | $28.620_{\pm 0.0003}$ |
|  | 10 | 35 | $\mathbf{14.390}_{\pm\mathbf{0.0899}}$ | $18.053_{\pm 0.0000}$ | $33.207_{\pm 0.0018}$ |
|  | 15 | 35 | $\mathbf{20.632}_{\pm\mathbf{0.5377}}$ | $24.875_{\pm 0.0000}$ | $34.532_{\pm 0.0008}$ |
|  | 20 | 35 | $\mathbf{28.140}_{\pm\mathbf{1.4211}}$ | $29.990_{\pm 0.0000}$ | $34.894_{\pm 0.0006}$ |
| Yelp | 10 | 25 | $\mathbf{9.6412}_{\pm\mathbf{0.0148}}$ | $11.640_{\pm 0.0000}$ | $23.112_{\pm 0.5788}$ |
|  | 10 | 30 | $\mathbf{9.8174}_{\pm\mathbf{0.0898}}$ | $12.587_{\pm 0.0000}$ | $27.396_{\pm 0.7311}$ |
|  | 10 | 35 | $\mathbf{10.008}_{\pm\mathbf{0.2310}}$ | $13.508_{\pm 0.0000}$ | $31.600_{\pm 0.9164}$ |
|  | 15 | 35 | $\mathbf{13.422}_{\pm\mathbf{0.0436}}$ | $18.237_{\pm 0.0000}$ | $33.257_{\pm 0.6701}$ |
|  | 20 | 35 | $\mathbf{18.160}_{\pm\mathbf{0.4387}}$ | $22.562_{\pm 0.0000}$ | $34.114_{\pm 0.4259}$ |

### 4.2.3 EFFECTIVENESS OF $L_e$ AND $L_n$

Training MTPP-CHD is achieved by minimizing loss $L_e$ and $L_n$. Minimizing $L_e$ is applied to force MTPP-CHD to move more events from $\boldsymbol{\mathcal{H}}_f$ to $\boldsymbol{\mathcal{H}}_d$ so that the two constraints in MTPP-CHD are satisfied. On the other hand, minimizing $L_n$ is applied to encourage MTPP-CHD to move fewer events from $\boldsymbol{\mathcal{H}}_f$ to $\boldsymbol{\mathcal{H}}_d$ so that $|\boldsymbol{\mathcal{H}}_d|$ is minimized. To verify that, Figure 3 (a) report the number of events in $\boldsymbol{\mathcal{H}}_d$ returned by the MTPP-CHD trained by minimizing $L_e$ only on dataset StackOverflow, and Figure 3 (b) report the number of events in $\boldsymbol{\mathcal{H}}_d$ returned by the MTPP-CHD trained by minimizing $L_n$ only on dataset StackOverflow. As expected, all events in $|\boldsymbol{\mathcal{H}}_f|$ are moved to $|\boldsymbol{\mathcal{H}}_d|$ in the former while no events in $|\boldsymbol{\mathcal{H}}_f|$ are moved to $|\boldsymbol{\mathcal{H}}_d|$ in the latter. The same results can be observed on other datasets in Appendix C.2).

Table 3: Total time used to solve all CHD tasks in test data (first three columns) and time used for MTPP-CHD training (last column).

| | MTPP-CHD | GS | RD | MTPP-CHD(Training) |
|---|---|---|---|---|
| StackOverflow ($|\mathbf{x}_o| = 15, |\mathcal{H}_f| = 40$) | **2.86h** | 33.4h | 27.2h | 24.3h |
| Retweet ($|\mathbf{x}_o| = 10, |\mathcal{H}_f| = 25$) | **9.09h** | 67.0h | 83.8h | 29.7h |
| Yelp ($|\mathbf{x}_o| = 10, |\mathcal{H}_f| = 25$) | **1.81h** | 13.5h | 17.0h | 14.3h |

### 4.2.4 MODEL EFFICIENCY

This section reports the performance of MTPP-CHD and baselines regarding time efficiency. For MTPP-CHD, it must be trained first to learn model parameters using training data and then solve CHD. For GS and RD, they are directly applied to solve CHD because they have no parameter to train. In Table 3, the first three columns report the total time of the trained MTPP-CHD and baselines to solve CHD on all $(\mathcal{H}_f, \boldsymbol{x}_o)$ pairs in three test datasets. More results are available in Appendix Table 9. The results tell that MTPP-CHD is significantly faster than baselines. The reason is that GS and RD have to interact with the MTPP model multiple times for one $\mathcal{H}_d$. On the other hand, the trained MTPP-CHD does not need to interact with MTPP model because it already learned which event should be distilled from MTPP during training. To have a better understanding of the time efficiency for MTPP-CHD, the last column of Table 3 reports the time used by MTPP-CHD for training (see Table 6 for training data size). It is comparable with the time consumed by GS and RD. Since MTPP-CHD only needs to be trained once, it is much more efficient compared with GS and RD.

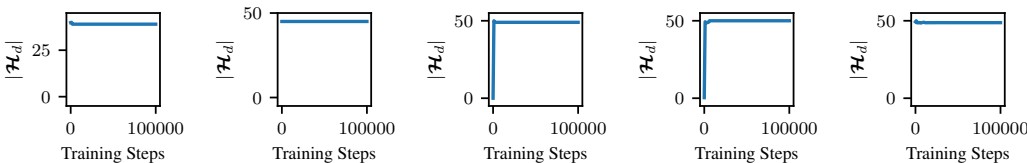

(a) The number of event in $\mathcal{H}_d$ returned by MTPP-CHD trained by minimizing $L_e$ only.

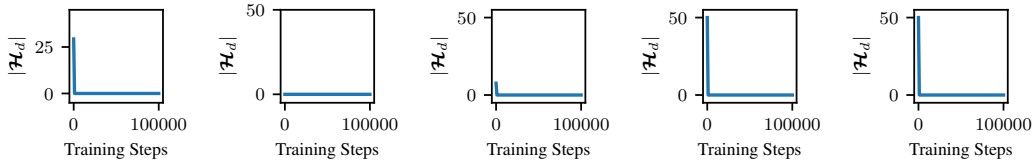

(b) The number of event in $\mathcal{H}_d$ returned by MTPP-CHD trained by minimizing $L_n$ only.

Figure 3: Effectiveness of $L_e$ and $L_n$ (from left to right: $(|\boldsymbol{x}_o|, |\mathcal{H}_f|) = (15, 40), (15, 45), (15, 50), (20, 50), (25, 50)$).

## 4.3 ANALYSIS OF DISTILLED EVENTS

The resultant $\mathcal{H}_d$ is a minimum subset of events in $\mathcal{H}_f$ that represents the essential information in history from the perspective of the underlying MTPP model. Specifically, the accuracy of MTPP model based on $\mathcal{H}_d$ is close to $\mathcal{H}_f$, and the accuracy of MTPP based on $\mathcal{H}_l$ is significantly lower than $\mathcal{H}_f$. Investigating the events in $\mathcal{H}_d$ may disclose interesting insights.

Given a dataset, the events with particular marks may influence the occurrence of the subsequent events more, for example, a retweet by famous users in Retweet. To verify it, we compare $\mathcal{H}_d$ returned using MTPP-CHD against $\mathcal{H}_d$ using RD on the test data of Retweet in terms of mark percentage. The mark percentage is calculated as the ratio of the number of events for that mark in $\mathcal{H}_d$s to the number of events for the same mark in $\mathcal{H}_f$s within the test data. RD randomly selects events from $\mathcal{H}_f$ to constitute $\mathcal{H}_d$. In contrast, $\mathcal{H}_d$ returned using MTPP-CHD has the essential information for predicting the next events. If a mark has more influence on the occurrence of the subsequent events, the mark is expected to be more frequent in $\mathcal{H}_d$ returned using MTPP-CHD than

using RD. From Figure 4, Mark 2 refers to famous users. We can observe that Mark 2 is consistently more frequent in $\mathcal{H}_d$ returned using MTPP-CHD while other marks are not. The result tells that the retweets by famous users have more influence on the occurrence of the subsequent retweets.

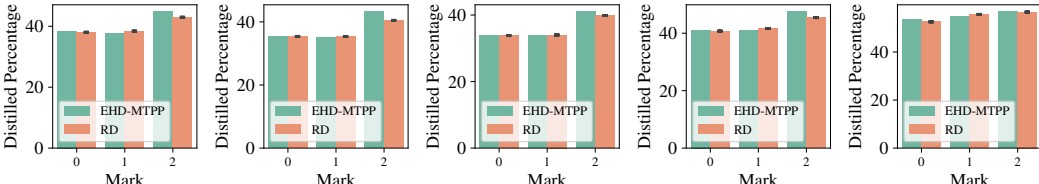

Figure 4: The percentage of events for different marks in $\mathcal{H}_d$ returned by MTPP-CHD and Random Distillation (RD) on test date of Retweet (from left to right: $(|\boldsymbol{x}_o|, |\mathcal{H}_f|) = (10, 25), (10, 30), (10, 35), (15, 35), (20, 35)$). All results pass the significance test with p-value 0.

## 5 RELATED WORKS

### 5.1 COUNTERFACTUAL ANALYSIS

**Counterfactual analysis on MTPP models** Recently, Noorbakhsh & Rodriguez (2022), Zhang et al. (2022b) and Hizli et al. (2023) used counterfactual analysis to investigate how the prediction of an MTPP model changes with handcrafted modifications of history. Noorbakhsh & Rodriguez (2022) successfully perform counterfactual analysis on the Hawkes process, an instance of MTPP, by deterministically accepting or rejecting the future events generated by the thinning algorithm (Ogata, 1981). Zhang et al. (2022b) use counterfactual analysis to estimate the influence of fake news engagements. By comparing the intensity function with manually modified history, they discover that users tend to behave differently if they recently engaged in misinformation. Hizli et al. (2023) use counterfactual analysis to evaluate the effect of medical treatments by checking how the blood glucose dynamics changes with and without a specific treatment.

CHD differs from existing counterfactual analysis related to MTPP models (Noorbakhsh & Rodriguez, 2022; Zhang et al., 2022b; Hizli et al., 2023). They investigate how a predefined modification to history would change the prediction of MTPP models. In contrast, CHD aims to find a minimal modification $\mathcal{H}_d$ so that the MTPP model can generate a distribution fitting $\boldsymbol{x}_o$ based on $\mathcal{H}_d$ but cannot based on $\mathcal{H}_l$. In summary, the methods in these studies cannot solve CHD.

**Counterfactual analysis on Classifiers** Some researchers use counterfactual analysis to analyze how binary and multi-class classifiers make decisions and name the task Counterfactual Explanations (CFE) (Verma et al., 2020). The definition of CFE involves a classifier $f$, an input feature $\mathbf{x}$, and an expected output $y$. We expect a counterfactual input $\mathbf{x}'$ by solving the following optimization problem:

$$\arg\min_{\mathbf{x}'} \quad d(\mathbf{x}, \mathbf{x}')$$
$$\text{s.t.} \quad f(\mathbf{x}') = y' \tag{13}$$

where $d(\mathbf{x}, \mathbf{x}')$ refers to the distance between $\mathbf{x}$ and $\mathbf{x}'$. Equation (13) means the expected $\mathbf{x}'$ should be similar to $\mathbf{x}$ while still changes the classification result from $y$ to $y'$. Usually, the similarity between $\mathbf{x}$ and $\mathbf{x}'$ means we should change as few features as possible, but sometimes it means the overall modification to $\mathbf{x}$ should be as small as possible (Verma et al., 2020). CFE generation is a well-investigated task with many existing works (Wachter et al., 2017; Dhurandhar et al., 2018; 2019; Joshi et al., 2019; Kanamori et al., 2020; Mothilal et al., 2020; Ramakrishnan et al., 2020; Parmentier & Vidal, 2021; Chen et al., 2022).

CHD is fundamentally different from CFE. CFE modifies the continuous input that would change the discrete output of a classifier (Verma et al., 2020). However, CHD manipulates the discrete input sequence that would change the continuous output of the MTPP model, *i.e.*, the accuracy for predicting the events observed later.

**Counterfactual analysis on Recommendation Systems** The recommendation system community has used counterfactual analysis to investigate how user behaviors and item features affect recommendation results (Mehrotra et al., 2018; Wang et al., 2020; Ghazimatin et al., 2020; Yang et al., 2021; Tran et al., 2021; Wang et al., 2021; Xu et al., 2021; Wang et al., 2022b; Zhong & Negre, 2022; Zhang et al., 2022b; Mu et al., 2022; Zhang et al., 2023a). Ghazimatin et al. (2020) proposed PRINCE, the first approach explaining recommendations concerning users' activities in Heterogeneous Information Networks(HIN). By greedily removing as few events as possible from the historical user event sequence that could replace the current recommendation with a different item, PRINCE identifies which interactions are responsible for model decisions. PRINCE heavily relies on the structure of HIN to efficiently find the solution, which limits its general use. To solve this, Tran et al. (2021) proposed ACCENT. It greedily searches for the smallest subset of history that the recommendation would change after training a new system with the subset removed. Zhong & Negre (2022) discuss applying SHAP(SHapley Additive exPlanations) (Lundberg & Lee, 2017) to greedily select features as the recommendation explanation. Zhang et al. (2023a) proposed PaGE-LINK. This graph-based explanation algorithm exploits the complete graph information from a learned GNN recommender to explain the recommendation results.

Besides, counterfactual analysis has been applied to understand how the reinforcement learning agent behaves in different environment states (Atrey et al., 2020; Wang et al., 2019; Li et al., 2021a; Zhou et al., 2022; Ji et al., 2023). Some researchers realize that they can detect and mitigate the bias in pretrained computer vision and language models by counterfactual analysis (Huang et al., 2020; Abbasnejad et al., 2020; Zhang et al., 2020c; Niu et al., 2021; Qian et al., 2021; Wang et al., 2022a).

## 5.2 Logic Point Processes

Besides counterfactual analysis, researchers have developed other ways to find causal relations between events on continuous time. One of the favorites is the Granger causality (Xu et al., 2016; Zhang et al., 2020b; Marcinkevics & Vogt, 2021; Zhu et al., 2022; Jalaldoust et al., 2022). Granger causality explores mutual relations across different marks, checking which mark helps the event forecast on other marks. Other works exploit logic rules from the temporal relation between different events, *e.g.*, one event happens before another event, then construct the conditional intensity function based on these relations (Li et al., 2021b; Yang et al., 2024; Song et al., 2024). Shi et al. (2023) use logic rules extracted by LLMs to improve the accuracy of next-event prediction. Zhang et al. (2021a) report an unsupervised approach to pick out exogenous events from a given sequence, called TPP-Select. TPP-Select separates all observed events into two types: endogenous events and exogenous events. Endogenous events occur because of historical influence, while exogenous events exist because of unknown external factors. By removing exogenous events from the dataset, TPP-Select can improve MTPP model training performance.

CHD differs from these works. CHD discloses causal relations between history and events observed later, while Granger causality (Idé et al., 2021; Wu et al., 2024) explores mutual relations across different marks to find which mark helps the event forecast on other marks. Other works (Li et al., 2020; Song et al., 2024) exploit logic rules between different events, *e.g.*, one event happens before another event, then construct the conditional intensity function based on these rules. In summary, the methods in these studies cannot solve CHD.

## 6 Conclusions

This study investigates Counterfactual History Distillation (CHD) to distill the essential events in history that can influence the occurrence of the subsequent events. This study demonstrates the issue of solving Counterfactual History Distillation (CHD) by conventional counterfactual analysis and refines the definition to ensure the distilled events are informative. With deliberate methods including Gumbel-softmax trick, the proposed solution MTPP-based Counterfactual History Distiller (MTPP-CHD) learns by effectively probing various event combinations. Its superiority has been observed in distillation optimization and processing speed in tests on real-world datasets. This study demonstrates analyzing the distilled events may disclose insights into the causal relation between events and event marks in continuous-time event sequences.

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

# A PROOFS

## A.1 PROOF OF PROPOSITION 1

*Proof.* CHD defined in Equation (7) has two constraints. For the first constraint, we have:

$$\log \text{ppl}(p(\boldsymbol{x}_o|\boldsymbol{\mathcal{H}}_f)) \leqslant \log \text{ppl}(p(\boldsymbol{x}_o|\boldsymbol{\mathcal{H}}_l)) + \log \epsilon_l \tag{14}$$

For the second constraint, we have:

$$\log \text{ppl}(p(\boldsymbol{x}_o|\boldsymbol{\mathcal{H}}_f)) \geqslant \log \text{ppl}(p(\boldsymbol{x}_o|\boldsymbol{\mathcal{H}}_d)) + \log \epsilon_d \tag{15}$$

By connecting Equation (14) and Equation (15), we get:

$$\frac{\text{ppl}(p(\boldsymbol{x}_o|\boldsymbol{\mathcal{H}}_l))}{\text{ppl}(p(\boldsymbol{x}_o|\boldsymbol{\mathcal{H}}_d))} \geqslant \frac{\epsilon_d}{\epsilon_l} \tag{16}$$

For any $\epsilon_l \in (0, 1)$ and $\epsilon_d \in (0, 1)$ where $\epsilon_d > \epsilon_l$, we can always move more events from $\boldsymbol{\mathcal{H}}_f$ to $\boldsymbol{\mathcal{H}}_d$ so that Equation (16) is satisfied. In the extreme case that $\frac{\epsilon_d}{\epsilon_l}$ is an any large number, all events in $\boldsymbol{\mathcal{H}}_f$ can be moved to $\boldsymbol{\mathcal{H}}_d$ so that $\boldsymbol{\mathcal{H}}_l = \varnothing$; then we have $\text{ppl}(p(\boldsymbol{x}_o|\boldsymbol{\mathcal{H}}_l)) \to +\infty$ that can always guarantee the inequation in Equation (16) held. □

# B EXPERIMENT DETAILS

## B.1 DATASETS

Table 4 reports the basic information of three real-world datasets, Retweet, StackOverflow, and Yelp. Table 5 shows different settings of $|\boldsymbol{\mathcal{H}}_f|$ and $|\boldsymbol{x}_o|$ for the subsequences $(\boldsymbol{\mathcal{H}}_f, \boldsymbol{x}_o)$ in experiments. Table 6 reports the number of events in training, validation, and test datasets for different settings of $|\boldsymbol{\mathcal{H}}_f|$ and $|\boldsymbol{x}_o|$. Table 7 presents the hyperparameters used for training the MTPP-CHD model on Retweet, StackOverflow, and Yelp. Because generating $\boldsymbol{\mathcal{H}}_d$ and $\boldsymbol{\mathcal{H}}_l$ from $\boldsymbol{\mathcal{H}}_f$ runs faster on the CPU, we train and evaluate all CHD approaches on Xeon Gold 6132 CPUs instead of GPUs.

**Retweet** (Zhao et al., 2015) records when users retweet a particular message on Twitter. The mark of this dataset distinguishes all users into 3 different types. Mark 0 refers to the normal user, whose follower count is lower than the overall median. Mark 1 refers to the influential user, whose follower count is higher than the median but lower than the top-5% of the entire user base. Mark 2 refers to the famous user, whose follower count is in the top-5% of the entire user base.

**StackOverflow** (Leskovec & Krevl, 2014) was collected from Stackoverflow[3], a popular question-answering website about various topics. Users providing decent answers will receive different badges as rewards. We have 22 marks in this dataset, representing 22 different badges that users can receive for their answers.

**Yelp**[4] contains the reviews of restaurants, shopping centers, and stores in the US on Yelp. We categorize these reviews into three groups based on the reviewers. Mark 0 refers to the normal reviewer. The number of reviews a normal reviewer has is lower than the overall median, which is 5 reviews in our case. Mark 1 refers to the influential reviewers. These reviewers write more reviews than normal reviewers but less than the top-5% reviewers. Mark 2 refers to the famous reviewers, the top-5% reviewers who write more than 92 reviews.

## B.2 MTPP MODEL

MTPP-CHD can work with any MTPP models that provide $p^*(m, t)$. Without loss of generality, this study uses FullyNN (Omi et al., 2019). Table 8 presents the hyperparameters used for training the FullyNN on Retweet, StackOverflow, and Yelp.

---

[3] https://stackoverflow.com
[4] https://www.yelp.com

Table 4: The basic information of datasets where the number of sequences, events, and marks are in the first three columns, $\bar{\tau}$ and $\sigma(\tau)$ are the mean and standard deviation of the time intervals between adjacent events, $t_0$ and $T$ are the earliest start time and the latest end time of all sequences.

| | Sequences | Events | Marks | $\bar{\tau}$ | $\sigma(\tau)$ | $t_0$ | $T$ |
|---|---|---|---|---|---|---|---|
| Retweet | 24 000 | 2 610 102 | 3 | 2574 | 16 302 | 0 | 604 799 |
| StackOverflow | 6633 | 480 414 | 22 | 0.8747 | 1.2091 | 1324 | 1390 |
| Yelp | 4022 | 409 946 | 3 | 7.2644 | 13.410 | 0 | 751 |

Table 5: Settings of $|\mathcal{H}_f|$ and $|\boldsymbol{x}_o|$ in experiments for each dataset.

| | (# of events in $\boldsymbol{x}_o$, # of events in $\mathcal{H}_f$) |
|---|---|
| Retweet | (10, 25), (10, 30), (10, 35), (15, 35), (20, 35) |
| StackOverflow | (15, 40), (15, 45), (15, 50), (20, 50), (25, 50) |
| Yelp | (10, 25), (10, 30), (10, 35), (15, 35), (20, 35) |

Table 6: The number of events in training, validation, and test dataset for different setting of $|\mathcal{H}_f|$ and $|\boldsymbol{x}_o|$.

| | $(\boldsymbol{x}_o, \mathcal{H}_f)$ | training | validation | test |
|---|---|---|---|---|
| Retweet | (10, 25) | 1 476 116 | 145 521 | 148 465 |
| | (10, 30) | 1 376 116 | 135 521 | 135 521 |
| | (10, 35) | 1 276 116 | 125 521 | 128 465 |
| | (15, 35) | 1 176 383 | 115 551 | 118 497 |
| | (20, 35) | 1 081 289 | 106 047 | 108 970 |
| StackOverflow | (15, 40) | 99 791 | 10 826 | 29 232 |
| | (15, 45) | 87 623 | 9451 | 25 824 |
| | (15, 50) | 77 341 | 8307 | 22 951 |
| | (20, 50) | 68 635 | 7350 | 20 504 |
| | (25, 50) | 61 254 | 6512 | 18 385 |
| Yelp | (10, 25) | 213 677 | 25 937 | 29 562 |
| | (10, 30) | 197 622 | 23 952 | 27 492 |
| | (10, 35) | 181 567 | 21 967 | 25 422 |
| | (15, 35) | 165 587 | 19 996 | 23 359 |
| | (20, 35) | 150 640 | 18 157 | 21 406 |

Table 7: Hyperparamters settings for training MTPP-CHD.

| | Retweet | StackOverflow | Yelp |
|---|---|---|---|
| Training Steps | 100 000 | 100 000 | 100 000 |
| Warmup Steps | 5000 | 5000 | 5000 |
| Batch Size | 256 | 128 | 128 |
| Hidden Vector | 64 | 64 | 64 |
| Input Vector | 32 | 32 | 32 |
| Q, K, V | 32 | 32 | 32 |
| Head | 4 | 4 | 4 |
| N | 4 | 4 | 4 |
| M | 4 | 4 | 4 |
| Learning Rate | 0.001 | 0.001 | 0.001 |
| $\epsilon_l$ | 0.5 | 0.5 | 0.6 |
| $\epsilon_d$ | 0.9 | 0.9 | 0.9 |
| $\alpha$ | 1.0 | 1.0 | 1.0 |
| $\beta$ | 1.0 | 1.0 | 1.0 |

FullyNN estimates the integral of conditional intensity functions $\Lambda^*(m, t) = \int_{t_l}^{t} \lambda^*(m, \tau)d\tau$ and calculates the value of the intensity function at time $t$ from the gradient of $\Lambda^*(m, t)$:

$$\Lambda^*(m,t) = \int_{t_l}^{t} \lambda^*(m,\tau)d\tau = \text{FullyNN}(m,t) \tag{17}$$

$$\lambda^*(m,t) = \frac{\partial \Lambda^*(m,t)}{\partial t} = \frac{\partial \text{FullyNN}(m,t)}{\partial t} \tag{18}$$

$$p^*(m,t) = \lambda^*(m,t)\exp\left(-\Lambda^*(t)\right) \tag{19}$$

$$= \frac{\partial \text{FullyNN}(m,t)}{\partial t}\exp\left(-\sum_{n\in\mathbb{M}}\text{FullyNN}(n,t)\right) \tag{20}$$

This helps FullyNN elude calculating $\Lambda^*(m,t)$ by numerical integration methods, such as Monte Carlo integration, to predict MTPP faster and more accurately. The FullyNN is trained on NVIDIA A100 GPUs.

Table 8: Hyperparamters settings for training MTPP Models.

|  | Retweet | StackOverflow | Yelp |
| --- | --- | --- | --- |
| Training Steps | 400 000 | 200 000 | 200 000 |
| Warmup Steps | 80 000 | 40 000 | 40 000 |
| Batch Size | 32 | 32 | 32 |
| History Embedding | 32 | 32 | 32 |
| Optimizer | AdamW | AdamW | AdamW |
| Intensity Vector | 16 | 32 | 16 |
| Learning Rate | 0.002 | 0.002 | 0.002 |
| Layers | 4 | 2 | 4 |

## C  ADDITIONAL EXPERIMENT RESULTS

Additional experiment results in Section 4.2 are reported here.

### C.1  EFFECTIVENESS OF COUNTERFACTUAL ANALYSIS REFINEMENT

Figure 5 demonstrates the distribution of $\log \text{ppl}(p(\boldsymbol{x}_o|\mathcal{H}_l)) - \log \text{ppl}(p(\boldsymbol{x}_o|\mathcal{H}_d))$ on StackOverflow, Retweet, and Yelp at various settings of $|\mathcal{H}_f|$ and $|\boldsymbol{x}_o|$ using MTPP-CHD with and without refinement, respectively. The results further support the conclusion in Section 4.2.1 that the resultant $\mathcal{H}_d$s have more information than the corresponding $\mathcal{H}_l$s for predicting the following events $|\boldsymbol{x}_o|$ using MTPP-CHD with refinement. In contrast, MTPP-CHD without refinement may lead to the resultant $\mathcal{H}_d$s having less information than the corresponding $\mathcal{H}_l$s.

### C.2  EFFECTIVENESS OF $L_e$ AND $L_n$

Section 4.2.3 demonstrate that minimizing loss $L_e$ leads to $\mathcal{H}_d$ with fewer events and minimizing loss $L_n$ leads to $\mathcal{H}_d$ with more events, respectively, on StackOverflow. The results on Retweet and Yelp are reported in Figure 6 and Figure 7, respectively. They are consistent with the results in Section 4.2.3.

### C.3  MODEL EFFICIENCY

Table 9 presents the total time of the trained MTPP-CHD and baselines to solve CHD on three real-world datasets at more settings of $|\mathcal{H}_f|$ and $|\boldsymbol{x}_o|$ in the first three columns, and the time used by MTPP-CHD for training on these datasets in the last column. The results futher demonstrate that MTPP-CHD solves CHD more efficiently than baselines.

### C.4  ANALYSIS OF DISTILLED EVENTS

In Figure 8 and Figure 9, we present the percentage of different marks in $\mathcal{H}_d$ returned by MTPP-CHD and RD on the test data of StackOverflow and Yelp. For StackOverflow, the results demonstrate some

marks have more information but others have less information for predicting the following events. For Yelp, all marks seemingly have the similar information about $\boldsymbol{x}_o$.

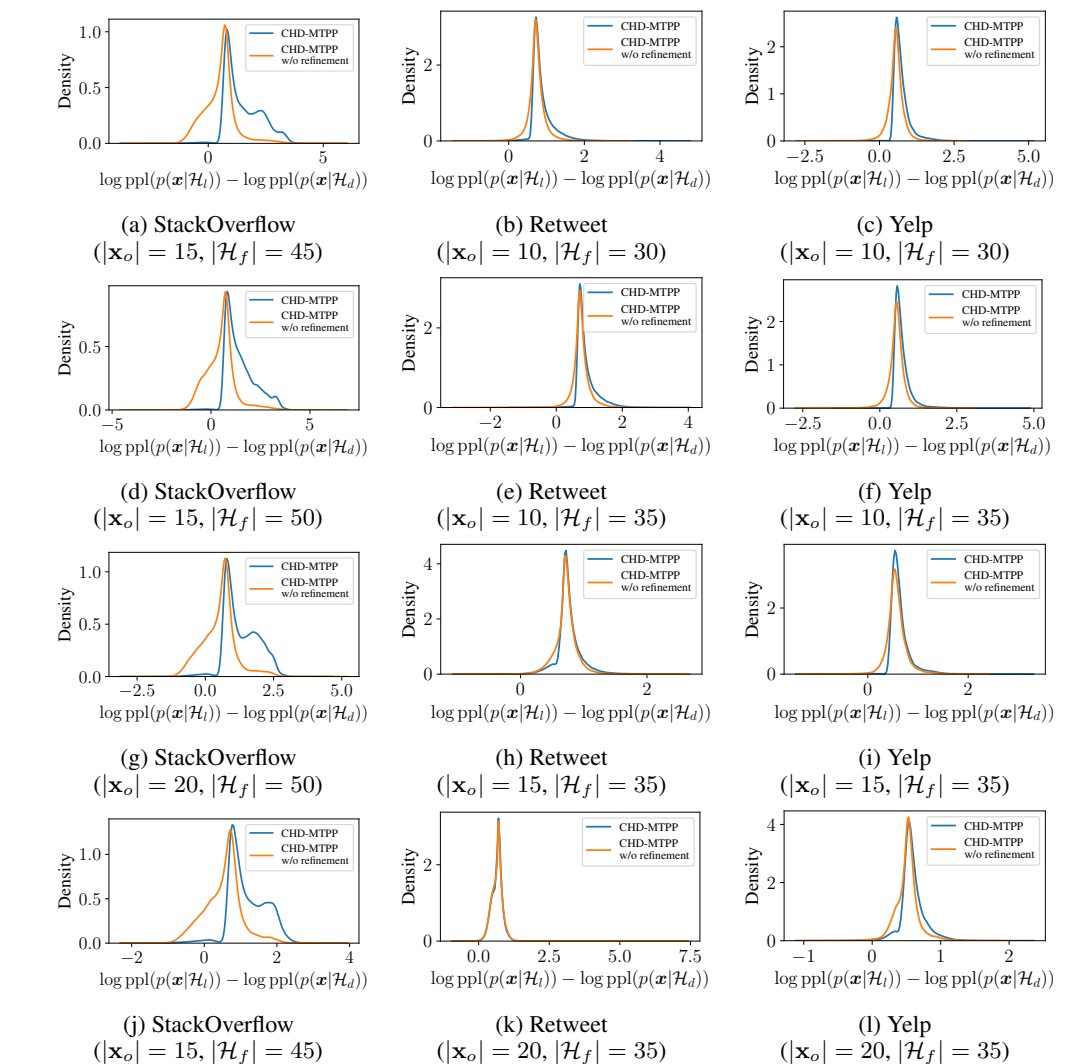

(a) StackOverflow
($|\mathbf{x}_o| = 15, |\mathcal{H}_f| = 45$)

(b) Retweet
($|\mathbf{x}_o| = 10, |\mathcal{H}_f| = 30$)

(c) Yelp
($|\mathbf{x}_o| = 10, |\mathcal{H}_f| = 30$)

(d) StackOverflow
($|\mathbf{x}_o| = 15, |\mathcal{H}_f| = 50$)

(e) Retweet
($|\mathbf{x}_o| = 10, |\mathcal{H}_f| = 35$)

(f) Yelp
($|\mathbf{x}_o| = 10, |\mathcal{H}_f| = 35$)

(g) StackOverflow
($|\mathbf{x}_o| = 20, |\mathcal{H}_f| = 50$)

(h) Retweet
($|\mathbf{x}_o| = 15, |\mathcal{H}_f| = 35$)

(i) Yelp
($|\mathbf{x}_o| = 15, |\mathcal{H}_f| = 35$)

(j) StackOverflow
($|\mathbf{x}_o| = 15, |\mathcal{H}_f| = 45$)

(k) Retweet
($|\mathbf{x}_o| = 20, |\mathcal{H}_f| = 35$)

(l) Yelp
($|\mathbf{x}_o| = 20, |\mathcal{H}_f| = 35$)

Figure 5: The distribution of $\log \mathrm{ppl}(p(\boldsymbol{x}_o|\boldsymbol{\mathcal{H}}_l)) - \log \mathrm{ppl}(p(\boldsymbol{x}_o|\boldsymbol{\mathcal{H}}_d))$ of our MTPP-CHD and the MTPP-CHD without refinement.

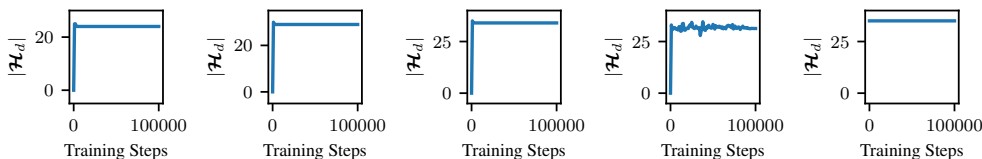

(a) The number of event in $\mathcal{H}_d$ returned by MTPP-CHD trained by minimizing $L_e$ only.

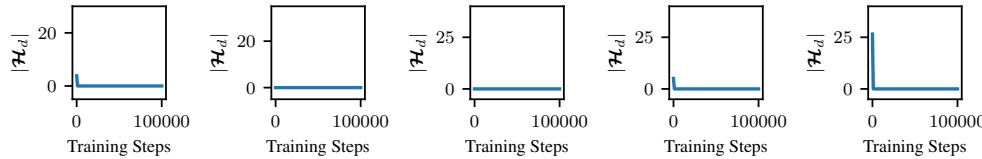

(b) The number of event in $\mathcal{H}_d$ returned by MTPP-CHD trained by minimizing $L_n$ only.

Figure 6: Effectiveness of $L_e$ and $L_n$ on Retweet (from left to right: $(|\boldsymbol{x}_o|, |\boldsymbol{\mathcal{H}}_f|) = (15, 40)$, $(15, 45)$, $(15, 50)$, $(20, 50)$, $(25, 50)$).

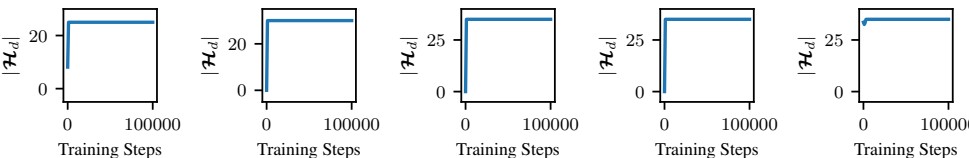

(a) The number of event in $\mathcal{H}_d$ returned by MTPP-CHD trained by minimizing $L_e$ only.

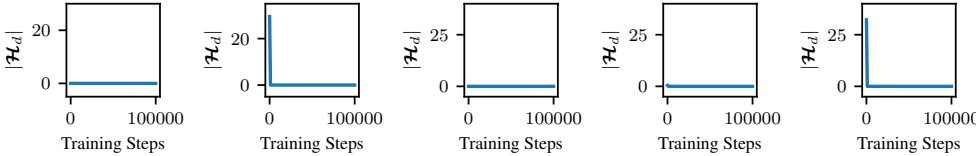

(b) The number of event in $\mathcal{H}_d$ returned by MTPP-CHD trained by minimizing $L_n$ only.

Figure 7: Effectiveness of $L_e$ and $L_n$ on Yelp (from left to right: $(|\boldsymbol{x}_o|, |\boldsymbol{\mathcal{H}}_f|) = (15, 40)$, $(15, 45)$, $(15, 50)$, $(20, 50)$, $(25, 50)$).

Table 9: Total time used to solve all CHD tasks in test data (first three columns) and time used for MTPP-CHD training (last column).

| | MTPP-CHD | GS | RD | MTPP-CHD(Training) |
|---|---|---|---|---|
| StackOverflow ($|\mathbf{x}_o| = 15, |\mathcal{H}_f| = 45$) | **2.83h** | 26.3h | 36.6h | 24.0h |
| StackOverflow ($|\mathbf{x}_o| = 15, |\mathcal{H}_f| = 50$) | **2.86h** | 26.7h | 41.1h | 25.3h |
| StackOverflow ($|\mathbf{x}_o| = 20, |\mathcal{H}_f| = 50$) | **2.58h** | 24.0h | 36.9h | 29.5h |
| StackOverflow ($|\mathbf{x}_o| = 25, |\mathcal{H}_f| = 50$) | **2.35h** | 21.7h | 33.4h | 31.8h |
| Retweet ($|\mathbf{x}_o| = 10, |\mathcal{H}_f| = 30$) | **9.93h** | 93.5h | 89.0h | 29.0h |
| Retweet ($|\mathbf{x}_o| = 10, |\mathcal{H}_f| = 35$) | **10.8h** | 102h | 111h | 33.4h |
| Retweet ($|\mathbf{x}_o| = 15, |\mathcal{H}_f| = 35$) | **10.1h** | 93.8h | 103h | 37.5h |
| Retweet ($|\mathbf{x}_o| = 20, |\mathcal{H}_f| = 35$) | **9.38h** | 87.5h | 94.9h | 36.8h |
| Yelp ($|\mathbf{x}_o| = 10, |\mathcal{H}_f| = 30$) | **2.01h** | 18.8h | 17.6h | 14.8h |
| Yelp ($|\mathbf{x}_o| = 10, |\mathcal{H}_f| = 35$) | **2.16h** | 20.3h | 22.1h | 15.4h |
| Yelp ($|\mathbf{x}_o| = 15, |\mathcal{H}_f| = 35$) | **2.00h** | 18.7h | 20.5h | 15.5h |
| Yelp ($|\mathbf{x}_o| = 20, |\mathcal{H}_f| = 35$) | **1.87h** | 17.2h | 18.9h | 15.2h |

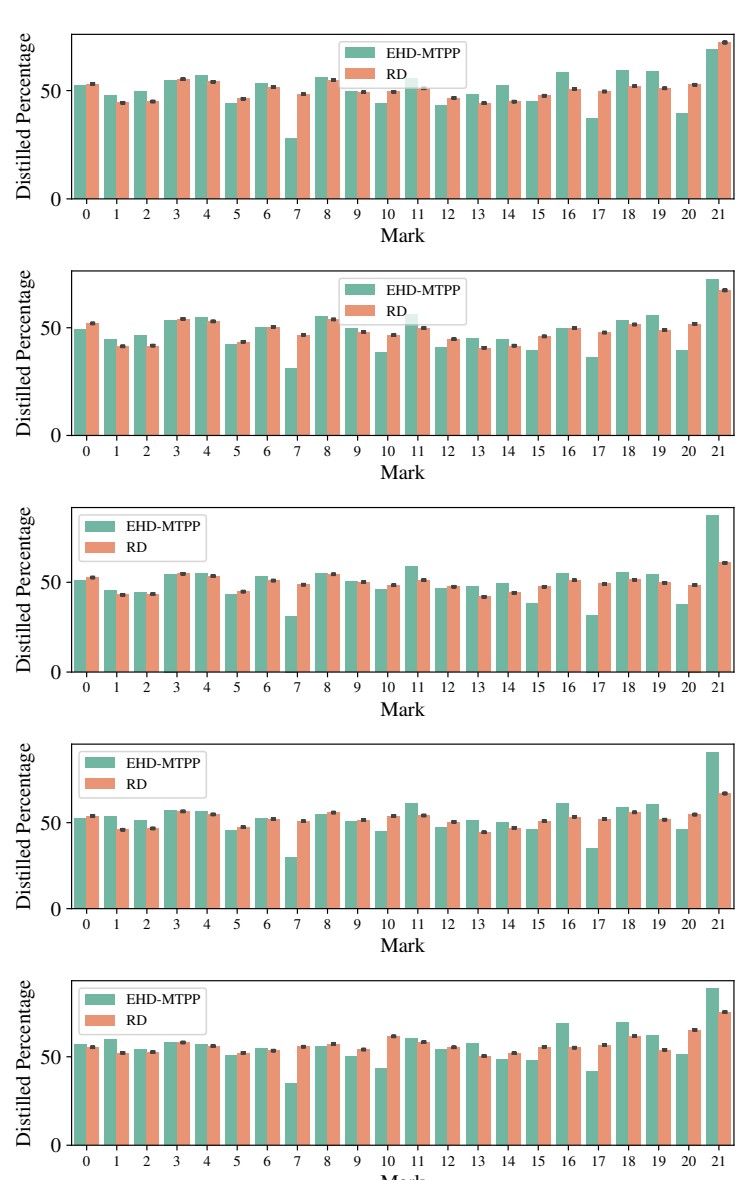

Figure 8: The percentage of events for different marks in $\mathcal{H}_d$ returned by MTPP-CHD and Random Distillation (RD) on test date of StackOverflow (from left to right: $(|\boldsymbol{x}_o|, |\mathcal{H}_f|) = (10, 25), (10, 30), (10, 35), (15, 35), (20, 35))$. The results pass the significance test with p-values smaller than $\alpha = 0.005$ for most marks.

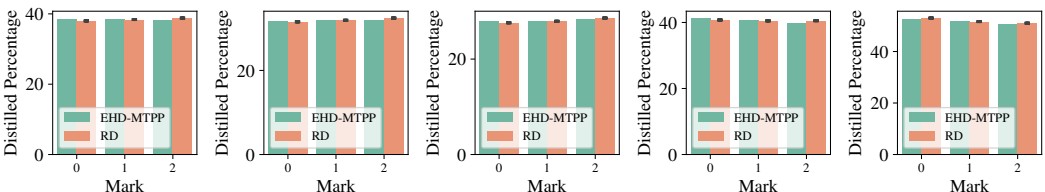

Figure 9: The percentage of events for different marks in $\mathcal{H}_d$ returned by MTPP-CHD and Random Distillation (RD) on test date of Yelp (from left to right: $(|\boldsymbol{x}_o|, |\mathcal{H}_f|) = (10, 25), (10, 30), (10, 35), (15, 35), (20, 35))$. The results pass the significance test with p-values smaller than $\alpha = 0.005$.

