# OpenReview forum: "Counterfactual History Distillation on Continuous-time Event Sequences"
_ICLR.cc/2025/Conference — ICLR 2025 Conference Withdrawn Submission_

### Official Review · Reviewer_fCu7 · 2024-10-24

**Soundness:** 2
**Presentation:** 3
**Contribution:** 3
**Rating:** 6
**Confidence:** 2

**Summary:**

This paper aims to distill history events that have essential information for predicting subsequent events with counterfactual analysis. They proposed a MTPP based counterfactual history distiller which learns to select the optimal event combination from history for the events observed later. Experiment results demonstrate the superiority of the proposed model in terms of distillation optimization and processing speed.

**Strengths:**

1. This paper raises an intriguing question: how to distill historical events containing crucial information for predicting future events in multivariate temporal point processes with counterfactual analysis. The research perspective is quite novel.
2. The related work section of the paper is comprehensive, offering insights into the distinctions between the proposed model and existing models. This demonstrates the author's in-depth exploration of the issue.
3. They demonstrate the issues when solving CHD by original counterfactual methods and propose one more constraint to refine it to ensure that the distilled events are desirable.
5. The experimental section is thorough, providing ample experimental details encompassing hardware and software requirements, as well as hyperparameter settings, ensuring reproducibility.

**Weaknesses:**

1. This paper provides the percentages of subsequences in three real-world datasets where ppl$(p(x_o|H_d))$ is greater than ppl$(p(x_o|H_l))$ by solving the optimization problem in Eq.(5). However, the percentages of Retweet and Yelp dataset are small. Moreover, seeing from Fig.2, especially for Retweet and Yelp dataset, the difference between results of MTPP-CHD and the results of MTPP-CHD w/o refinement is small. I am concerned about the prevalence of this phenomenon in real-world datasets and whether its occurrence is linked to the presence of anomalous events. Is it possible to employ anomaly detection methods for an initial screening of the dataset, potentially reducing the necessity of the additional constraints proposed in the paper?

2. The experiments were solely conducted on real-world datasets, and validating the results on synthetic datasets would enhance the model's credibility. I recommend the authors refer to [1,2] to generate synthetic datasets using temporal logic rules and then assess whether the model proposed in this paper can distill key events effectively.

3. The authors claim that no previous studies investigated CHD in the context of MTPP and there are no suitable baselines. But the baselines of GS and RD are too naive, which could result in a lack of persuasive efficiency comparisons with these two baseline models. Could the authors explore additional baseline models?

4. The analysis in Sec 4.3 makes sense, since in the temporal point process, there may be only the occurrence of some certain types of events that can impact the intensity of events in subsequence. But the author only provides the results of the influence by different types of events on the Retweet dataset. I suggest the author provide similar analysis and report the results using tables or figures for the key events distilled in Tab.2, which will enhance the persuasiveness of the model's effectiveness.

5. How is the proposed model in the paper adapted for common prediction tasks in temporal point processes?

**Questions:**

My concerns align with the points outlined in the weakness section:

1. Does the prevalence of the phenomenon in real-world datasets as shown in Tab.1 frequent? Does its occurrence link to the presence of anomalous events? Is it possible to employ anomaly detection methods for an initial screening of the dataset, potentially reducing the necessity of the additional constraints proposed in the paper?

2. Experiments on synthetic dataset are needed. I recommend the authors refer to [1, 2] to generate synthetic datasets using temporal logic rules and then assess whether the model proposed in this paper can distill key events effectively.

3. Lack of ablation study. The authors can compare the impact of different MTPP models on estimating the conditional probability distribution, assess how varying top-k selections in the historical event picker stage influence model performance, and evaluate the effect of introducing additional constraints on the model's effectiveness.

4. Could the authors explore additional baseline models?

5. Could the authors provide similar analysis as in Fig. 4 and report the results using tables or figures for the key events distilled in Tab. 2, which will enhance the persuasiveness of the model's effectiveness.

6. How is the proposed model in the paper adapted for common prediction tasks in temporal point processes?

reference:\
[1] Temporal Logic Point Processes\
[2] Explaining Point Processes by Learning Interpretable Temporal Logic Rules

---

### Official Review · Reviewer_22Up · 2024-10-31

**Soundness:** 2
**Presentation:** 3
**Contribution:** 2
**Rating:** 3
**Confidence:** 2

**Summary:**

The authors introduced Counterfactual History Distillation (CHD) for Neural marked TPP which identifies the essential subset of historical events needed to accurately predict future occurrences using a Marked Temporal Point Process (MTPP) model.  By leveraging counterfactual analysis, CHD determines the smallest set of events (Hd) whose removal significantly decreases prediction accuracy for future events (xo), with the remaining events forming Hl. The authors present MTPP-CHD, a method that uses an encoder-decoder transformer to select events for Hd and Hl. Experiments on Retweet, StackOverflow, and Yelp datasets show that MTPP-CHD outperforms baseline methods in both quality and speed, producing shorter Hd sets that meet perplexity constraints.

**Strengths:**

Originality: the authors proposed a new concept Counterfactual History Distillation (CHD) for Neural marked TPP. There is no work has been done before.

Quality: the optimization approach differentiate from other modeling approaches (as the authors mentioned in the 3 papers, counterfactual TPPs, Causal inference for TPPs, and counterfactual outcome under a different policy which I am very familiar.) Authors motivate well, proposes the approach reasonably, and conduct thorough experiments.

Clarity: it is well presented for the most part; I can follow fairly well.

Significance: I think this is an interest line of research and worthy of investigation by our community.

**Weaknesses:**

1.	I think the major concern/ weakness is the context or tools for counterfactual analysis. The authors mentioned the 3 papers (or the 4 additional by hizli et al); the underlying framework is either potential outcome by robbins/rubin or causal structure models by pearl. However I cannot find how the authors draw connection with either framework. I am also trying to justify by potentially using causally disentanglement where we have h_f and we predict x_o, and under a different setting h_d and we predict counterfactual x_o. but I also did not see any reasonable explanation.

2.	Another concern is the evaluation: as the authors mentioned
“Counterfactual analysis reveals casual relations by searching for the smallest modification to the input that could completely change the output “ I think I am more convinced if I can see the event subsequence x_o under h_d in your experiments visually, although the authors shows in figure 2 the modeling fitting density (or difference).  I also think the causal relations here is in the sense of Granger causality.

3.	My third concern is the model architecture in Figure 1 specially the history distiller. Maybe I am unaware of the knowledge distillation literature; but I am not sure how such encoder and decoder distillates history? And the generated sequences of binary conditional distribution is good for sampling influencing events in H_d and H_l. The authors did not explain section 3.

**Questions:**

1.	The author should give a better definition on H which will be the stamped events.

2.	Also the setting needs to be defined. H_f are given, and x_o is tested online? In eqn 4, suppose we x_o = {x_1,x_2,x_3}, p(x_1 | H) = p(x_1| H_f) and p(x_2| Hf and x_1)? Is this correct?

3.	How is the hyperparameters alpha and beta selected? Should beta be large enough to enforce the constraint?

4.	How good is the solution from Eqn 7?  Proposition 1 does not give us more information.

5.	Markov condition experiments. I think about a potential experiment for the authors to do: consider a case where the events are markovian such that p(x_i+1 | H) = p(x_i+1 | x_i). So that we can generate synthetic sequences, and we can find a groundtruth, that is only the last event in H_f matters. Is that a reasonable experiment?

6.	Baseline Models. The authors can improve by adding another baseline : recent history, selecting only most recent |H_d| events and see the outcome subsequences.  Using recent history is not uncommon for example see Gao, Tian, et al. "Causal inference for event pairs in multivariate point processes." NeurIPS 21.

---

### Official Review · Reviewer_xSjS · 2024-11-01

**Soundness:** 3
**Presentation:** 3
**Contribution:** 3
**Rating:** 3
**Confidence:** 4

**Summary:**

This paper investigate the history distillation problem for predicting subsequent events. To characterize the fitness of the selected events, this paper resort to perplexity metric. Based on the perplexity metric, a problem definition considering the information of subset and the subset size  is proposed. The authors deliver a method called MTPP-based Counterfactual History Distiller (MTPP-CHD) to solve it. Experimental results unveil that the selected subset contains information to predict the occurance of the subsequent events. And the proposed method is much more time efficient than baselines.

**Strengths:**

This paper propose a new problem which is disregard by previous works. The experiemental results analyze the comprehensive aspects of the method and the problem.

**Weaknesses:**

I have the following concerns about this paper.

1. Although the studied is a new problem which has not been investigated before, the importance of the problem is not clearly discussed. I suggest to supplement more discussion on the potential application of this technology in the introduction.

2. The effectiveness of the method seems not significant in the experiment section. The authors only consider the method with CHD-MTPP w/o refinement as the baselines in the comparison of the informativeness (i.e. Figure 2). The improvement shown in Figure 4 is also minor. And the authors did not compare the CHD-MTPP with GS.

**Questions:**

1. In Line 175, the authors define $p(x_o|\varnothing)$ as an infinitesimal number. I am confused why this is true. Can the integral of $p(x_o|\varnothing)$ be guaranteed to be 1?

2. The problem name is Counterfactual History Distillation. I am not sure whether it is related to counterfactual concept in causal inference. The counterfactual question the result if an imagined intervention is conducted. However, in this paper, it seems nothing is relevant to intervention.

---

### Official Review · Reviewer_cmpy · 2024-11-02

**Soundness:** 1
**Presentation:** 2
**Contribution:** 1
**Rating:** 3
**Confidence:** 2

**Summary:**

Summary

The goal is to distill history events to have the required info for predicting subsequent events with counterfactual analysis where events in history may have a causal relationship with the events observed later;

The authors propose MTPP Counterfactual History Distiller (MTPP-CHD), which learns to select the optimal event combination from history for the events observed later.

It is expected distilled events would have essential information but empirical results show otherwise; to address limitation, they add one more constraint to enforce that distilled events are informative;

Authors claim:
a) to distill a minimum subset of history events with the essential information for predicting the subsequent events using MTPP;
b) Show CHD limitations on classical counterfactual approach and add constraint;
c) proposed MTPP_CHD with gumbel-softmax trick;

Contributions can be splitted into three components:
1) History distiller, encoder-decoder transformer, p(y|Hf,x0)
2) Historical event picker
3) Training loss

**Strengths:**

Appreciate the addition of the algorithms and the description of the empirical evaluation

**Weaknesses:**

- Abstract is a bit confusing and the overall text was hard to follow. Notation was not very intuitive. The indexes on section 2 were very confusing.

- One of the conclusions presented was “The result tells that the retweets by famous users have more influence on the occurrence of the subsequent retweets. “(4.3), which is a known result since early 2010s.

[1] Suh, Bongwon, et al. "Want to be retweeted? large scale analytics on factors impacting retweet in twitter network." 2010 IEEE second international conference on social computing. IEEE, 2010.

[2] Rosenman, Evan TR. Retweets—but not just retweets: Quantifying and predicting influence on twitter. Diss. Doctoral dissertation, Bachelor’s thesis, applied mathematics. Harvard College, Cambridge, 2012.

Can the authors clarify how their method adds or differs from those previous findings?

- Granger causality, despite the misleading name, doesn’t actually find ‘causal relations’ as claimed on section 5.2. It is more of a statistical test to check if two time-series are related.

- One of the main challenges in counterfactual evaluation is the lack of ground truth, i.e., either a famous user retweeted or not, they can do both. While one can ‘mimic’ or ‘simulate’ what might have happened otherwise, there are several chain-effects that could jeopardize the results, and it is unclear how the authors addressed this one the paper. Example: how can one completely remove the retweet effects of a famous user from the events?  So could the authors explain how they handle the lack of ground truth in counterfactual scenarios and how they account for potential chain effects when removing certain events from the history.

**Questions:**

I had some questions on "weakness sections". Here are a few more questions:

Q1) As previously mentioned, evaluation of counterfactual scenarios using real-world data (i.e., tweets) is often a hard problem due to lack of ground-truth. While the causality community has used distributions to simulate the counterfactuals, these often required strong assumptions. Can the authors expand/explain how they ensured the ‘counterfactual’ scenario adopted is actually realistic to evaluate the method.

Q2) Why aren’t the indexes sequential? H_t_i represents history up to time t_l, and t_i< t_j if i<j. (Section 2.1).  Then in row 106, the first part ends on (x_j) is denoted H_f. Why not H_j? Row 132, there is H_d, H_f, H_l, which are also confusing. Could the authors provide a clear explanation of their notation choices and maybe revise the text for consistency if there's no specific reason for the current choices.

Q3) There were some mentions of counterfactual analysis on Recommender systems. Considering RL also handles sequential observations, could one of these methods be used as baseline?

Q4) Can the authors revise their language in Section 5.2 to more accurately describe what Granger causality measures.

---

### Note · Authors · 2025-01-17

I have read and agree with the venue's withdrawal policy on behalf of myself and my co-authors.